# A shotgun metagenomic investigation of the microbiota of udder cleft dermatitis in comparison to healthy skin in dairy cows

Lisa Ekman[1,2]*, Elisabeth Bagge[1], Ann Nyman[2,3], Karin Persson Waller[1,2], Märit Pringle[1], Bo Segerman[4,5]

**1** Department of Animal Health and Antimicrobial Strategies, National Veterinary Institute, Uppsala, Sweden, **2** Department of Clinical Sciences, Swedish University of Agricultural Sciences, Uppsala, Sweden, **3** Växa Sverige, Stockholm, Sweden, **4** Department of Microbiology, National Veterinary Institute, Uppsala, Sweden, **5** Department of Medical Biochemistry and Microbiology, Uppsala University, Uppsala, Sweden

\* lisa.ekman@slu.se

**Data Availability Statement:** The raw sequence data has been submitted to the Sequence Read Archive (SRA) and is accessible via the bioproject

## Abstract

Udder cleft dermatitis (UCD) is a skin condition affecting the fore udder attachment of dairy cows. UCD may be defined as mild (eczematous skin changes) or severe (open wounds, large skin changes). Our aims were to compare the microbiota of mild and severe UCD lesions with the microbiota of healthy skin from the fore udder attachment of control cows, and to investigate whether mastitis-causing pathogens are present in UCD lesions. Samples were obtained from cows in six dairy herds. In total, 36 UCD samples categorized as mild (n = 17) or severe (n = 19) and 13 control samples were sequenced using a shotgun metagenomic approach and the reads were taxonomically classified based on their *k*-mer content. The Wilcoxon rank sum test was used to compare the abundance of different taxa between different sample types, as well as to compare the bacterial diversity between samples. A high proportion of bacteria was seen in all samples. Control samples had a higher proportion of archaeal reads, whereas most samples had low proportions of fungi, protozoa and viruses. The bacterial microbiota differed between controls and mild and severe UCD samples in both composition and diversity. Subgroups of UCD samples were visible, characterized by increased proportion of one or a few bacterial genera or species, e.g. *Corynebacterium*, *Staphylococcus*, *Brevibacterium luteolum*, *Trueperella pyogenes* and *Fusobacterium necrophorum*. *Bifidobacterium* spp. were more common in controls compared to UCD samples. The bacterial diversity was higher in controls compared to UCD samples. Bacteria commonly associated with mastitis were uncommon. In conclusion, a dysbiosis of the microbiota of mild and severe UCD samples was seen, characterized by decreased diversity and an increased proportion of certain bacteria. There was no evidence of a specific pathogen causing UCD or that UCD lesions are important reservoirs for mastitis-causing bacteria.

PRJNA636289. All other relevant data are within the paper and its Supporting Information files.

**Funding:** K.P.W. received funding from The Swedish Research council Formas (grant number 221 -2013-269, www.formas.se) and from Stiftelsen lantbruksforskning - Swedish farmers' foundation for agricultural research (grant number V1430006, www.lantbruksforskning.se). The funders had no role in study design, data collection and analysis, decision to publish, or preparation of the manuscript. A.N. is employed by Växa Sverige, but her salary costs for her work in the study was covered by the grants listed above. Thus, Växa Sverige did not have any role in the study design, data collection and analysis, decision to publish, or preparation of the manuscript. The specific roles of these authors are articulated in the 'author contributions' section.

**Competing interests:** The authors have declared that no competing interests exist. A.N. is employed by the commercial company Växa Sverige. This does not alter our adherence to PLOS ONE policies on sharing data and materials.

## Introduction

Udder cleft dermatitis (UCD) is a skin condition that affects the anterior parts of the udder in dairy cows. It has been reported in the UK [1], the USA [2], Sweden [3, 4], Denmark [5], the Netherlands [6] and Norway [7]. The prevalence varies between studies, but in high-prevalence herds, up to 60% of cows may be affected [8]. The UCD lesions vary in appearance and may be classified as mild or severe, based on whether or not skin integrity is breached [4, 8, 9]. The etiology and pathogenesis of the lesions are still largely unknown. Recent studies indicate a multifactorial origin of UCD, associated with both cow- and herd-related risk factors, such as parity, breed, udder conformation, high herd-level production and type of floor in cubicles [4, 6, 8, 9]. In addition, several infectious agents have been implicated in the development of UCD, such as mange mites [10], *Treponema* spp. [11] and Bovine herpesvirus 4 [12], but the true role of these agents in the etiology of UCD has not been proven. Moreover, culturing of swab samples from UCD lesions has revealed a variety of aerobic and anaerobic bacteria, as well as fungi [2, 3, 13], indicating that the lesions may be a reservoir for pathogens, potentially increasing the risk of infectious diseases such as mastitis. In line with this, a few studies have found associations between UCD, particularly severe cases, and an increased risk of clinical mastitis [4, 14], but it is not known whether mastitis-causing pathogens are a common finding in UCD microbiota. Previous microbiological investigations of UCD lesions have mainly been performed through culturing [3, 13], microscopy [13] or *Treponema*-specific PCR assays [11, 15]. In recent decades, the use of culture-independent methods to identify the microorganisms present in a sample or an environment has become increasingly common [16]. So far, few studies have been performed on samples from UCD lesions, although a recent study used 16S rRNA-amplicon sequencing to investigate the bacterial microbiota of UCD lesions and compared it with that of healthy skin [17]. They found that certain bacterial genera were more common in samples from UCD lesions, such as *Fusobacterium*, *Helcococcus*, *Anaerococcus*, *Trueperella* and *Porphyromonas*, compared to samples from healthy skin. In the 16S rRNA-amplicon sequencing method, specific regions the rRNA gene is PCR amplified and sequenced of from bacteria, to assess the microbiota [18]. Shotgun metagenomic sequencing is an alternative method to analyze the microbiota in which total DNA is sequenced using only a limited number of amplification cycles and this method can detect all types of microbes with improved resolution down to the species and strain level [19, 20]. This method has been used in studies on human gut [21] and bovine ruminal [22] microbiota, as well as in studies on human skin microbiota, for example, in patients with atopic dermatitis [23], and the microbiota of human chronic wounds, such as pressure wounds and venous leg ulcers [24]. We believe that shotgun metagenomic sequencing has the potential to yield additional information on the microbiota of UCD lesions and increase the understanding of the development and clinical course of UCD and give indications how to treat these lesions.

Thus, the main objective of this study was to compare the microbiota of recently developed mild and severe UCD lesions, and healthy skin at the same body site using shotgun metagenomic sequencing to investigate whether specific microbes are associated with UCD lesions. We also wanted to investigate whether common mastitis-causing pathogens are present in UCD lesions, which would indicate that UCD may be a reservoir for udder infections.

## Material and methods

### Study design and participating cows

Seven Swedish dairy herds with free-stall housing and milking parlors were enrolled in the study. Inclusion criteria were a previous UCD prevalence of 20–40% [8] and that they were

located within 200 km of Uppsala, Sweden. The mean herd size was 125 cows (range 87–168 cows), mean herd level production was 10,204 kg milk/cow and year (range 7,680–11,534 kg) and the most common breeds were Swedish Red and Swedish Holstein. Herd visits were performed regularly from April 2018 to April 2019 as part of a longitudinal study of UCD (nine visits per herd at six-week intervals). This study design made it possible to identify and sample cows with recently developed UCD lesions. Ethical approval for this study was issued by a regional Swedish Ethics committee (appointed by the Swedish Board of Agriculture). The herd visits were conducted during one milking and all milking cows were scored for UCD (no, mild or severe). All scoring and sampling were performed by a single researcher. Mild UCD was defined as erythema and small papules or pustules, or small crusts, and severe UCD was defined as a breach of skin integrity, often with large crusts and exudative or bleeding wounds (Fig 1). Cows for sampling were chosen based on their UCD status. The criteria for sampling was a cow with a previous status of no UCD that received a score of mild or severe UCD, as well as a cow with a previous status of mild UCD that received a score of severe UCD. For every cow with a sampled UCD lesion, the aim was to sample the skin from the same body site (fore udder attachment and between the front quarters) of a control cow with no UCD. As far as possible, control cows of the same breed and parity as the UCD cows were selected. At the final herd visit, samples were also obtained from cows with previously registered UCD lesions in order to achieve a total of approximately 10 samples per category (no UCD, mild UCD and severe UCD) from each herd. Thus, cows that had been previously sampled could be sampled again at the final herd visit.

## Sampling procedure

Sampling was performed in the milking parlor, during milking or just after the milking unit had been removed. Clean disposable gloves were used at all samplings and were changed between cows. If the area for sampling (Fig 1) was visibly dirty, it was cleaned with paper (dry or soaked in water) or sterile gauze compresses (dry or soaked in saline 0.9%, Fresenius Kabi, Bad Homborg, Germany). Severe UCD lesions were always cleaned with sterile gauze compresses soaked in saline to remove loose crusts, necrotic tissue and pus. Finally, the area for sampling was wiped with one dry sterile gauze compress just before sampling. This step was

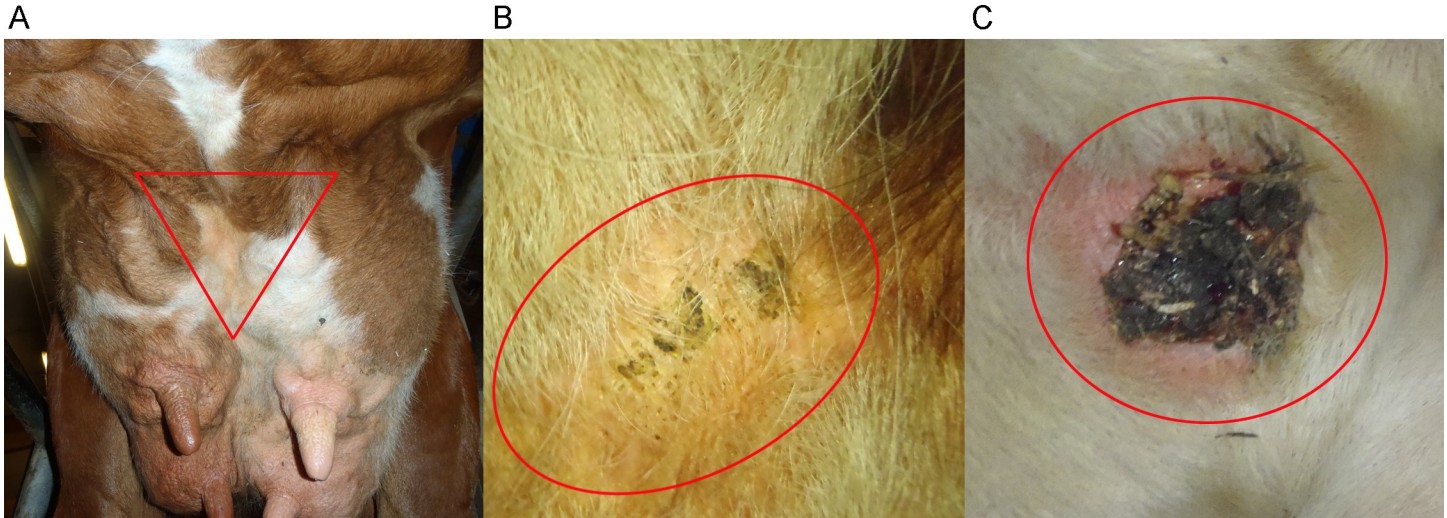

**Fig 1. Illustration of sampling site.** Samples were taken from (A) healthy control skin at the fore udder attachment, (B) mild and (C) severe udder cleft dermatitis.

also performed before sampling mild UCD lesions and healthy skin. Each sample was taken using a 50 cm² sponge moistened with saline contained in a sterile Minigrip bag (TS/15-B: NACL, Technical Service Consultants Ltd, Lancashire, UK) according to the manufacturer's instructions. The area for sampling was wiped using approximately 20 strokes, covering the entire lesion and adjacent skin (approximately 1–5 cm of skin around the lesion, depending on lesion size) or the ventral mid-area of the fore udder attachment and the area between the front quarters for healthy skin samples (Fig 1). Samples were uniquely labeled and immediately put on ice. They were kept cold (at 4˚C) during transportation and arrived at the laboratory (Uppsala, Sweden) within 24 hours. A total of 184 samples were taken from cows with no (n = 77), mild (n = 46) or severe (n = 61) UCD. As one herd had very few cases of UCD, the samples from this herd (n = 5) were excluded, leaving 179 samples from 6 herds for further analyses.

## Sampling analyses

The samples were processed within a few hours of arrival at the laboratory. First, 50 ml of sterile 0.9% saline (SVA, Uppsala, Sweden) was poured into the Minigrip bag. In order to dislodge microorganisms from the sponge into the fluid, the bag was treated in a stomacher (230 rpm; Stomacher® 400 Circulator, Seward, West Sussex, UK) for two minutes. The fluid was then poured into a sterile 50 ml plastic tube (Sarstedt, Nümbrecht, Germany) and the tube was centrifuged for 15 minutes at 2,000 g. Most of the supernatant was removed, leaving around 1–2 cm of fluid at the bottom of the tube and the pellet was dissolved in the remaining fluid (approximately 2–5 ml). The solution was then transferred into a 2 ml sterile plastic microtube (Sarstedt, Nümbrecht, Germany) and the samples were kept frozen at -23˚C for 1–8 weeks before DNA extraction.

**DNA extraction.** The microtube samples were thawed at room temperature for 20–40 minutes, briefly vortexed and then centrifuged for two minutes at 2,000 g. The supernatant was removed and the pellet was used for DNA extraction using the DNeasy Powerlyzer Powersoil Kit (12855–100, Qiagen AB, Sollentuna, Sweden) according to the manufacturer's instructions and with the following additions: solution C1 and solution C6 were heated to 65˚C before use to avoid precipitation and the samples were heated to 100˚C before the bead-beating step to improve the lysis of cellular structures. The bead-beating step was performed using a FastPrep -24™ homogenizer (MP Biomedicals, Irvine, CA, USA), with the settings 6.5m/s and MP24x2, for 2x60 seconds. After the extraction, the DNA concentration of each sample was measured by fluorometry using Qubit™ 1X dsDNA HS Assay Kit (Q33230, Thermo Fisher Scientific, Waltham, MA, USA) and varied between 0 and 110 ng/μl. The extracted DNA was stored at -23˚C until sequenced. From each herd and category (no, mild and severe UCD), 5–6 samples with sufficient DNA concentration were chosen for further analyses–a total of 96 samples. At the sequencing facility (SNP&SEQ Technology Platform, Uppsala, Sweden), the DNA concentration was re-measured with Quant-iT™ (Thermo Fisher Scientific) and DNA fragmentation was analyzed with an Agilent Fragment Analyzer (DNF-467-kit, Santa Clara, CA, USA). Some samples had a high degree of DNA fragmentation. We therefore chose 49 samples with acceptable quality parameters for sequencing, 13 from healthy skin (controls), 17 from mild UCD lesions and 19 from severe lesions.

**DNA sequencing.** Sequencing libraries were prepared from 10 ng of DNA using the SMARTer ThruPLEX DNA-Seq kit (R400676, Takara-Clontech, Saint-Germain-en-Laye, France) according to the manufacturer's preparation guide #080818. Briefly, the DNA was fragmented using a Covaris E220 system (Covaris Inc, Woburn, MA, USA), aiming at 400 bp fragments. The ends of the fragments were end-repaired and stem-loop adapters were ligated

to the 5' ends of the fragments. The 3' end of the stem loop was subsequently extended to close the nick. Finally, the fragments were amplified and unique index sequences were introduced using seven cycles of PCR followed by purification using AMPure XP beads (Beckman Coulter Inc., Indianapolis, IN, USA). The quality of the library was evaluated using the Agilent Fragment Analyzer system (DNF-910-kit). The adapter-ligated fragments were quantified by qPCR using the Library Quantification Kit for Illumina (KAPA Biosystems/Roche, Wilmington, MA, USA) on a CFX384 Touch instrument (BioRad, Hercules, CA, USA) prior to cluster generation and sequencing. A 400 pM pool of the sequencing libraries in an equimolar ratio was subjected to cluster generation and paired-end sequencing with a 150bp read length in a SP flowcell and the NovaSeq6000 system (Illumina Inc., San Diego, CA, USA), using the v1 chemistry according to the manufacturer's protocols. Base calling was performed on the instrument by RTA 3.3.4 and the resulting.bcl files were demultiplexed and converted to fastq format with tools provided by Illumina Inc., allowing for one mismatch in the index sequence. Additional statistics on sequence quality were compiled with an in-house script from the fastq files, RTA and CASAVA output files. Sequencing was performed by the SNP&SEQ Technology Platform (Uppsala, Sweden). The raw sequence data has been submitted to the Sequence Read Archive (SRA) and is accessible via the bioproject PRJNA636289. The SRA accessions are listed in S1 Table.

**Bioinformatic analyses.** The fastq files were first trimmed using Trimmomatic [25]. The parameters for Trimmomatic were "SE -threads 6 ILLUMINACLIP:adaptes.fa:2:30:10 LEADING:3 TRAILING:3 SLIDINGWINDOW:4:15 MINLEN:36 X.fastq.gz X.trimmed.fastq.gz". To remove contaminating cow sequences, the fastq files were then mapped to the *Bos taurus* genome (ARS-UCD1.2) with Bowtie2 [26] using standard settings. The mapped and unmapped reads were separated using Samtools [27]. Only paired reads where both were unmapped to *Bos taurus* were kept. A Kraken2 database was built (Sep 2020) with Archaea, Bacteria, fungi, protozoa, viral and UniVec Core sequences according to the instructions in the manual, and used with Kraken2 [28]. The parameters for Kraken2 classifications were "—db krakendb—threads 10—paired X_R1.fastq X_R2.fastq—report X.krakenreport.txt". The Kraken results were then run through Bracken [29], to estimate Species, Genera and Phylum level data. The parameters for bracken-build were "-d krakendb -t 10 -k 35 -l 150" and for Bracken "bracken -d krakendb -i X.krakenreport.txt -o X.bracken.txt -r 150 -l (S or G or P)". The results were visualized using Pavian, a web application for exploring metagenomics classification results [30]. Some of the severe samples showed pronounced elevated levels of the intracellular parasite *Babesia*, which infects red blood cells. There was also a correlation between the number of *Babesia* reads and the number of reads mapped to the cow genome in the same sample. Given the association of *Babesia* with red blood cells and the correlation to cow DNA, the *Babesia* reads were deemed as contamination due to blood in the sample and were excluded from the analysis.

## Statistical analyses

Custom Perl scripts were created to merge Kraken report files into a single table with clade counts for each sample. Counts were expressed as a percentage of all classified reads identified as Bacteria, Archaea, Eukarya (i.e. fungi or protozoa) or virus, and were analyzed descriptively and compared between groups using the Wilcoxon rank sum test. A data dimensionality reduction with principal component analysis (PCA) was performed on the bacterial phylum, genus and species level. Bacterial phyla, genera and species that represented at least 10% in at least one sample were analyzed for differences between control samples and mild and severe UCD samples, respectively, using Wilcoxon rank sum tests and Bonferroni correction to adjust for multiple comparisons. In addition, Fisher's exact test was used to investigate the

distribution of herd, breed and parity between the three groups. The alpha diversity of bacterial species and genera was investigated by calculating Shannon diversity indexes for all samples using $H\prime = -\sum_{i=1}^{R} p_i \ln p_i$, where $p$ = proportional abundance of each taxa. The diversity was compared between sample types using Wilcoxon rank sum tests. Bacterial species considered to be common mastitis-causing pathogens in Sweden [31] (listed in S2 Table) that represented more than 1% of the classified reads in at least one sample were described and differences in abundance were compared between control samples and mild and severe UCD samples using the Wilcoxon rank sum test. Archaeal phyla and genera were investigated descriptively and by PCA. Fungal reads were investigated descriptively. Two cows were sampled at two different time points. One of these cows was sampled with a mild lesion, and then sampled again when the lesion became severe. The other cow was sampled once when it had a recently developed severe lesion, and then again at the last sampling, three months later, when the lesion was still severe. These samples were investigated descriptively as they could yield information on how the microbiota of UCD lesions can change over time. Statistical calculations were performed using Stata (release 15.1; StataCorp LLC, College Station, TX, USA) and PCA and heatmaps were generated using the TM4 Multiple experiment Viewer (MEV), version 4.8.

## Results

An overview of the sequenced samples including cow information, UCD category, quality control results, numbers of sequenced reads and the proportion of reads that mapped to the bovine genome is shown in S1 Table and detailed results from the classification by Kraken2 and Bracken in S2 Table. There were no associations between herd, breed or parity and sample type ($P$ = 0.96, $P$ = 0.42 and $P$ = 0.23, respectively).

### Overall microbial abundance

Reads that could be classified to the domains of Bacteria, Archaea and Eukarya and to viruses were used for an analysis of the microbiota composition. Eukarya was further divided into fungal and protozoan groups. All samples were strongly dominated by reads from the Bacteria domain, except for one sample (M6), which had almost 40% fungal reads (Fig 2A). However, apart from this deviating sample, most samples had low (less than 1%) proportions of fungi. The proportion of Archaea classified reads was lower in samples from mild and severe UCD (means 2.8 (SD 1.5) and 0.5% (SD 0.4), respectively) compared to control samples (mean 4.1% (SD 1.0); Fig 2B). The viral reads and the protozoan reads, after filtering out reads from the red blood cell parasite *Babesia*, each constituted less than 1% of the reads within all samples (Fig 2B) and there was no clear pattern of differences between UCD lesions and control samples. For these reasons, the protozoan and viral reads were not investigated further.

### Bacteria

The unsupervised data dimensionality reduction with PCA on the phylum, genus and species level revealed different subgroups of UCD samples (Fig 3). The major driving taxa affecting the PC axes are shown in S1 Fig. On the phylum level (Fig 3A), the first PCA axis (PC1) separated a large subgroup of both mild and severe UCD samples. Both the second (PC2) and third (PC3) PCA axis separated other largely non-overlapping smaller subgroups of UCD samples, one only including severe (PC2) and one mainly including mild (PC3) UCD samples. The third PCA axis also separated the majority of the control samples, together with a few of the mild UCD samples. Most, but not all, UCD samples were separated from the control samples on at least one of the three first PCA axes.

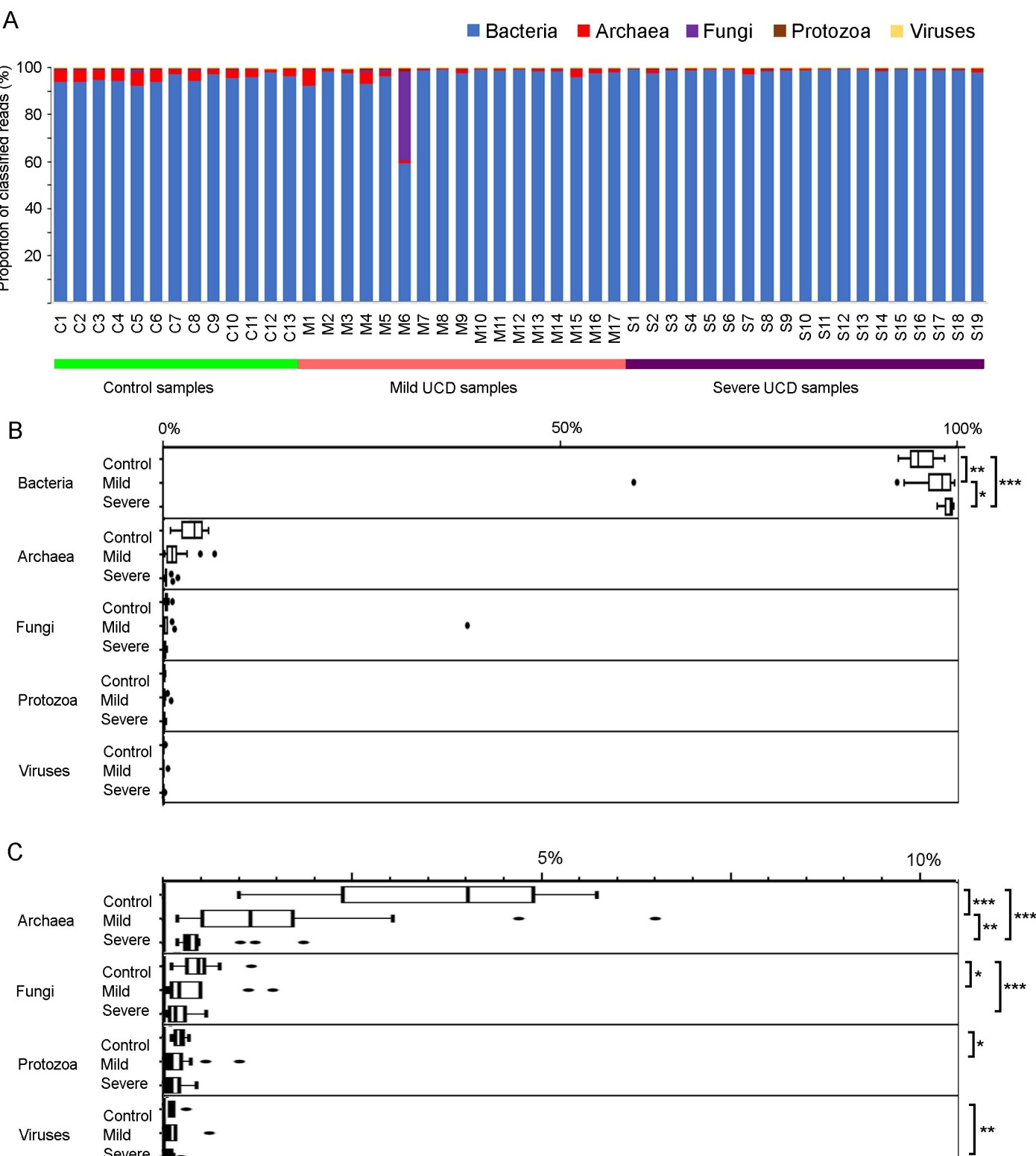

**Fig 2. Microbial abundance.** Distribution of reads classified as Bacteria, Archaea, fungi, protozoa and virus within each sample (A) and sample type (B and C) based on samples from mild (M, n = 17) and severe (S, n = 19) udder cleft dermatitis (UCD) and samples from skin at the fore udder attachment from healthy controls (C, n = 13). The *P*-values from comparing the abundance between sample types are denoted in the margins of B and C by *** if $P \leq 0.001$, ** if $P \leq 0.01$ and * if $P \leq 0.05$. The exact *P*-values were for Bacteria: C/M $P = 0.01$, C/S $P < 0.0001$, M/S $P = 0.02$, Archaea: C/M $P = 0.0007$, C/S $P < 0.0001$, M/S $P = 0.004$, fungi: C/M $P = 0.05$, C/S $P = 0.0007$, protozoa: C/M $P = 0.04$, and viruses: C/M $P = 0.2$, C/S $P = 0.002$.

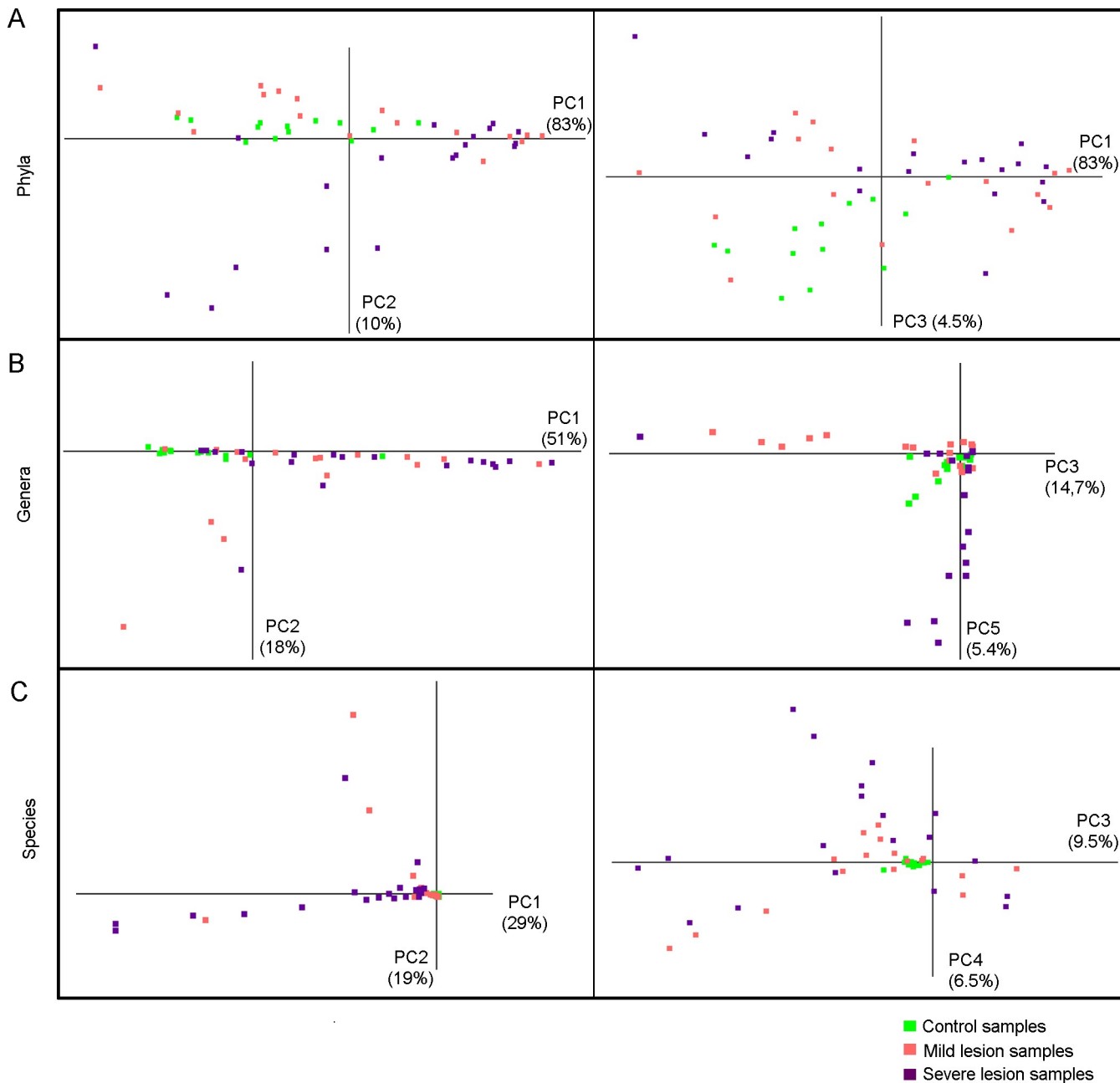

**Fig 3. Unsupervised analysis.** Data dimensionality reduction with principal component analysis (PCA) performed on the bacterial phylum (A), genus (B) and species level (C) for 49 samples from mild (n = 17) and severe (n = 19) udder cleft dermatitis and samples from skin at the fore udder attachment from healthy controls (n = 13). The (%) given for each PCA axis indicates the variation explained by that specific axis. The fourth PCA axis on genus level separated a subgroup of control samples and is presented in S2 Fig.

On the genus level, the first PCA axis separated a large group of both mild and severe UCD samples including one control sample (C12), while the other PCA axes separated smaller subgroups, mainly including mild or severe UCD samples (Fig 3B). The healthy control samples were more strongly clustered than the UCD samples, except for C12 at PC1, and C9 and C13 at PC 4 and 5 (Fig 3B and S2 Fig). On the species level, the control samples were tightly clustered close to the origin, while the UCD samples were separated into several different

subgroups along the PCA axes (Fig 3C). Overall, the PCA analysis suggested that most of the UCD samples were distinctly separable from the control samples, although there appeared to be more than one subgroup of UCD samples. We found no indication that the clustering of samples were related to herd, breed or parity of the animal, or that the extraction month had an association with the results (S3 Fig).

**Bacterial abundance.** The control samples and the mild UCD samples were dominated by three phyla, Actinobacteria, Firmicutes and Proteobacteria (Fig 4A). Although there was no overall difference between sample types, a substantial number of both mild and severe UCD samples showed a markedly higher proportion (around 80% or higher) of Actinobacteria compared to control samples. The group of samples with a high proportion of Actinobacteria corresponded to the samples separated by the first PCA axis on the phylum level (Fig 3A). A subgroup of the severe UCD samples had a markedly high proportion of the phyla Fusobacteria and Bacteroidetes compared to other samples (Fig 4A). This subgroup largely corresponded to the samples separated by the second PCA axis on the phylum level (Fig 3A). In the most pronounced cases, around one third of the bacterial reads belonged to Fusobacteria or Bacteroidetes. Another group of samples, mainly from mild UCD, was reflected in the third PCA axis on the phylum level and had a relatively high proportion of Firmicutes compared to other UCD samples. The third PCA axis also separated the majority of the control samples, together with a few of the samples from mild UCD and in general, a higher proportion of Proteobacteria was seen in control samples compared to mild and severe UCD samples (Table 1).

On the genus and species level, the taxa that represented more than 10% in at least one sample were visualized (Fig 4B and 4C), and differences in abundance of these taxa between sample types are presented in Table 1. In many UCD samples, a single genus represented a larger proportion of the reads compared to control samples (Fig 4B). The specific genus and species that had increased in proportion differed between samples, but some subgroups were visible. The subgroups were also defined by hierarchical clustering (S4 Fig). The largest subgroup had a high proportion of *Corynebacterium* spp. and corresponded to the samples separated by the first PCA axis on the genus level (Fig 3B and S1 Fig). In addition, this was largely the same group of samples that was dominated by Actinobacteria on the phylum level. Different *Corynebacterium* species dominated in different samples but, in most cases, one or two species represented a major proportion (>50%) of the *Corynebacterium* associated reads within each sample (Fig 4C). The first PCA axis on species level separated a group of mainly severe UCD samples with a high proportion of *Corynebacterium lactis*, whereas *Corynebacterium urealyticum*, *C. xerosis* and *C. camporealensis* contributed to the separation of several mild and severe UCD samples by the third PCA axis (PC3, Fig 3C and S1 Fig). Several of these *Corynebacterium* spp. differed significantly between sample types (Table 1). A few samples had a higher proportion of *Brevibacterium*, mainly *Brevibacterium luteolum* (Fig 4B and 4C), which corresponded to the second PCA axis on the genus and species level (Fig 3B and 3C and S1 Fig). In a third subgroup, mainly including mild UCD lesions, an increased proportion of *Staphylococcus* spp. was seen (Fig 4B and 4C). This group corresponds to the samples separated by the third PCA axis on the genus level (Fig 3B and S1 Fig) and largely represents the group in which Firmicutes had expanded on the phylum level (Fig 3A). The fourth identified subgroup only comprised severe UCD samples and had a high proportion of anaerobic or facultative anaerobic bacteria, including the genera *Trueperella*, *Fusobacterium* and *Porphyromonas* (Fig 4B and 4C). This group was separated on the fifth PCA axis on the genus level and largely corresponded to the group characterized by Fusobacteria and Bacteroidetes on the phylum level (Fig 3B and S1 Fig). The most dominating species in this group was *Porphyromonas asaccharolytica*, *Trueperella pyogenes*, and *Fusobacterium necrophorum* and the latter two also affected the fourth PCA axis on species level (Fig 3C and S1 Fig). Several control samples had a

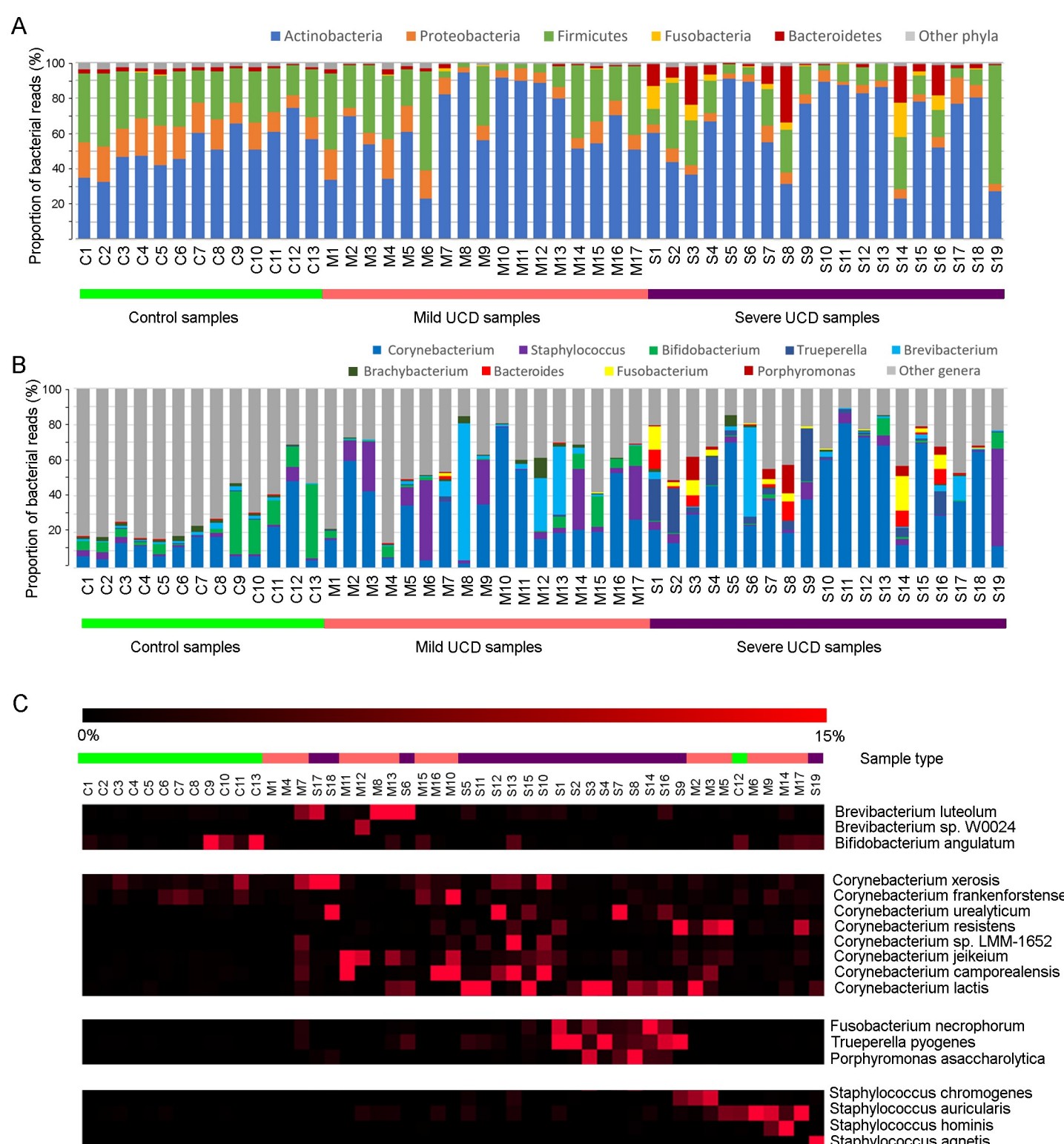

**Fig 4. Bacterial abundance.** Distribution of bacterial phyla (A), genera (B) and species (C) representing ≥10% of the classified reads in at least one sample out of 49 samples from mild (n = 17) and severe (n = 19) udder cleft dermatitis (UCD) lesions and skin samples from healthy controls (n = 13) in a study of the microbiota of UCD in comparison to healthy skin using shotgun metagenomic sequencing. On the species level (C), the order of the samples was changed to highlight the major subgroups that were distinguishable, and the red colour indicates the percentage (0–15%) of the bacterial reads for each species within each sample.

**Table 1. Comparison of UCD samples and healthy skin.**

| Rank[a] | Taxa | Control | Mild | *P* (M) | Severe | *P* (S) |
|---|---|---|---|---|---|---|
| P | Bacteroidetes | 2.07 | 0.72 | *0.001* | 2.26 | *0.54* |
| G | *Bacteroides* | 0.22 | 0.07 | *0.04* | 0.50 | *0.60* |
| G | *Porphyromonas* | 0.04 | 0.17 | *0.07* | 0.75 | *0.01* |
| S | *Porphyromonas asaccharolytica* | 0.02 | 0.01 | *0.06* | 0.54 | *0.006* |
| P | Fusobacteria | 0.16 | 0.07 | *0.03* | 0.86 | *0.009* |
| G | *Fusobacterium* | 0.10 | 0.04 | *0.03* | 0.83 | *0.005* |
| S | *Fusobacterium necrophorum* | 0.03 | 0.01 | *0.03* | 0.78 | *0.002* |
| P | Proteobacteria | 17.27 | 8.23 | *0.002* | 5.03 | *<0.0001* |
| P | Actinobacteria | 50.55 | 60.78 | *0.07* | 77.04 | *0.06* |
| G | *Brachybacterium* | 1.43 | 0.58 | *0.23* | 0.17 | *0.0007* |
| G | *Brevibacterium* | 1.18 | 1.19 | *0.90* | 0.62 | *0.45* |
| S | *Brevibacterium luteolum* | 0.36 | 0.25 | *0.80* | 0.35 | *0.79* |
| S | *Brevibacterium sp. W0024* | 0.02 | 0.02 | *0.74* | 0.02 | *0.25* |
| G | *Bifidobacterium* | 5.52 | 1.51 | *0.02* | 0.32 | *0.0001* |
| S | *Bifidobacterium angulatum* | 1.01 | 0.11 | *0.03* | 0.05 | *0.002* |
| G | *Corynebacterium* | 11.68 | 27.06 | *0.02* | 37.96 | *0.0001* |
| S | *Corynebacterium camporealensis* | 0.33 | 0.50 | *0.10* | 0.60 | *0.02* |
| S | *Corynebacterium frankenforstense* | 1.18 | 0.63 | *0.46* | 0.33 | *0.04* |
| S | *Corynebacterium jeikeium* | 0.20 | 1.25 | *0.0007* | 0.62 | *<0.0001* |
| S | *Corynebacterium lactis* | 0.23 | 0.25 | *0.28* | 4.27 | *<0.0001* |
| S | *Corynebacterium sp. LMM-1652* | 0.05 | 0.24 | *0.001* | 0.17 | *<0.0001* |
| S | *Corynebacterium resistens* | 0.03 | 0.38 | *<0.0001* | 0.25 | *<0.0001* |
| S | *Corynebacterium urealyticum* | 0.10 | 0.24 | *0.04* | 0.77 | *<0.0001* |
| S | *Corynebacterium xerosis* | 1.10 | 0.84 | *0.26* | 1.69 | *0.88* |
| G | *Trueperella* | 0.17 | 0.12 | *0.17* | 2.87 | *<0.0001* |
| S | *Trueperella pyogenes* | 0.13 | 0.10 | *0.28* | 2.87 | *<0.0001* |
| P | Firmicutes | 27.31 | 23.99 | *0.68* | 10.47 | *0.002* |
| G | *Staphylococcus* | 1.40 | 2.86 | *0.14* | 1.08 | *0.91* |
| S | *Staphylococcus agnetis* | 0.005 | 0.008 | *0.21* | 0.007 | *0.27* |
| S | *Staphylococcus auricularis* | 0.12 | 1.28 | *0.18* | 0.11 | *0.57* |
| S | *Staphylococcus capitis* | 0.03 | 0.07 | *0.54* | 0.04 | *0.29* |
| S | *Staphylococcus chromogenes* | 0.01 | 0.03 | *0.32* | 0.04 | *0.36* |
| S | *Staphylococcus hominis* | 0.05 | 0.06 | *0.93* | 0.02 | *0.03* |

Bacterial phyla, genera and species representing at least 10% of the classified reads in at least one sample of samples from mild (M, n = 17) and severe (S, n = 19) udder cleft dermatitis (UCD) and 13 control (C) samples from cows without UCD. The 49 samples were obtained from 47 cows in six Swedish dairy herds. The median proportion of classified reads for each bacteria and sample type is presented. Differences in abundance between control samples and mild (M) UCD, and between control samples and severe (S) UCD samples were analyzed using the Wilcoxon rank sum test. A P-value ≤0.002 was considered significant due to multiple testing according to the Bonferroni correction.

[a]P = Phylum, G = Genus, S = Species.

relatively high proportion of *Bifidobacterium spp.* and these samples were separated by the fourth PCA axis on genus level (Fig 4C and S2 Fig). The abundance of several genera and species distinguishing these subgroups differed significantly between UCD categories and control samples (Table 1).

**Bacterial diversity.** On the genus level, the mean Shannon diversity index was significantly higher in controls compared to UCD samples, with a mean of 4.7 (SD 0.8) for control samples, 3.3 (SD 1.0) for mild UCD samples and 2.8 (SD 0.8) for severe UCD samples (Fig

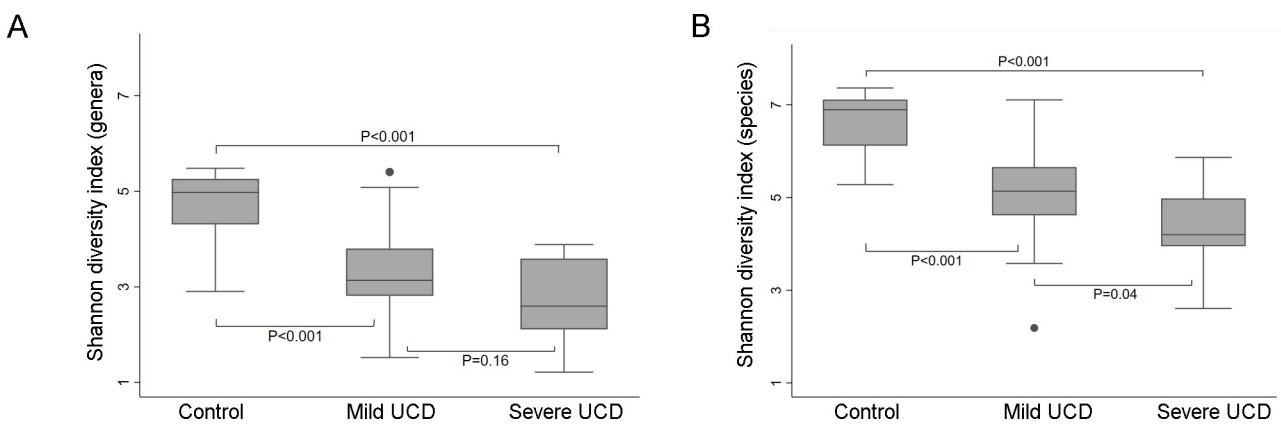

**Fig 5. Bacterial diversity.** Box plot of the Shannon diversity index for bacterial genera (A) and species (B), in samples from mild (n = 17) and severe (n = 19) udder cleft dermatitis (UCD) and samples from skin at the fore udder attachment from healthy controls (n = 13).

5A). Also, on the species level, there was significantly higher diversity in control samples (mean 6.6, SD 0.7) compared to mild (mean 5.1, SD 1.2) and severe (mean 4.3, SD 0.8)) UCD samples, as well as a higher diversity in mild compared to severe UCD samples (Fig 5B).

 **Mastitis-causing bacteria and spirochetes in UCD samples.** Among the pathogens that are considered to be common mastitis-causing bacterial species in Sweden, *Escherichia coli*, *Staphylococcus chromogenes*, *Staph. epidermidis*, *Staph. haemolyticus*, *Staph. simulans* and *Trueperella pyogenes* were represented by at least 1% of the reads in at least one sample. There was a higher proportion of *Escherichia coli* in the control samples compared to mild ($P$ = 0.006) and severe ($P<0.0001$) UCD samples, although the proportion generally was low in all sample types. For *Staph. epidermidis* there was a tendency towards higher proportions in the control samples ($P$ = 0.02). For the other *Staphylococcus* spp. mentioned above, no differences between sample types were seen. *Trueperella pyogenes* was more frequent in samples from severe lesions (mean 7.1%, SD 9.3, $P<0.0001$), $P$ = 0.0001) compared to samples from control cows (mean 0.1, SD 0.06), whereas there was no significant difference between controls and mild UCD samples (mean 0.3. SD 0.5, $P$ = 0.28). The Spirochaetes proportion of the classified reads ranged from 0.01 to 0.4% (mean 0.2%) and *Treponema* spp. reads constituted between 0.002 and 0.2% of the classified reads, with a very low abundance in all sample types.

### Other microorganisms

 **Archaea.** As concluded above, there was a lower proportion of sequence reads classified as Archaea in UCD samples compared to control samples ($P$ = 0.007 for mild and $P<0.0001$ for severe UCD samples; Figs 2B and 6A). Most of the severe UCD samples and a subset of the mild UCD samples had a very low proportion of archaeal classified reads, i.e. less than 0.5% (Fig 6A). These samples were also different in the relative composition of archaeal genera (Fig 6B). In the control samples and some of the UCD samples, reads classified to the genus *Methanobrevibacter* dominated. In the other UCD samples, two subgroups were distinguishable. One group, including both mild and severe UCD samples, was characterized by a high relative proportion of halophilic archaea (e.g. *Halorobrum* and *Halobacterium*) and the other, including severe UCD samples only, by an increased proportion of *Methanosarcina* (Fig 6B). The *Methanosarcina* subgroup was largely the same as the subgroup characterized by elevated levels of anaerobic bacteria described above. A PCA on the genus level separated samples with low abundance of archaea on the first PCA axis and the halophilic and *Methanosarcina* subgroups on the second PCA axis (Fig 6C).

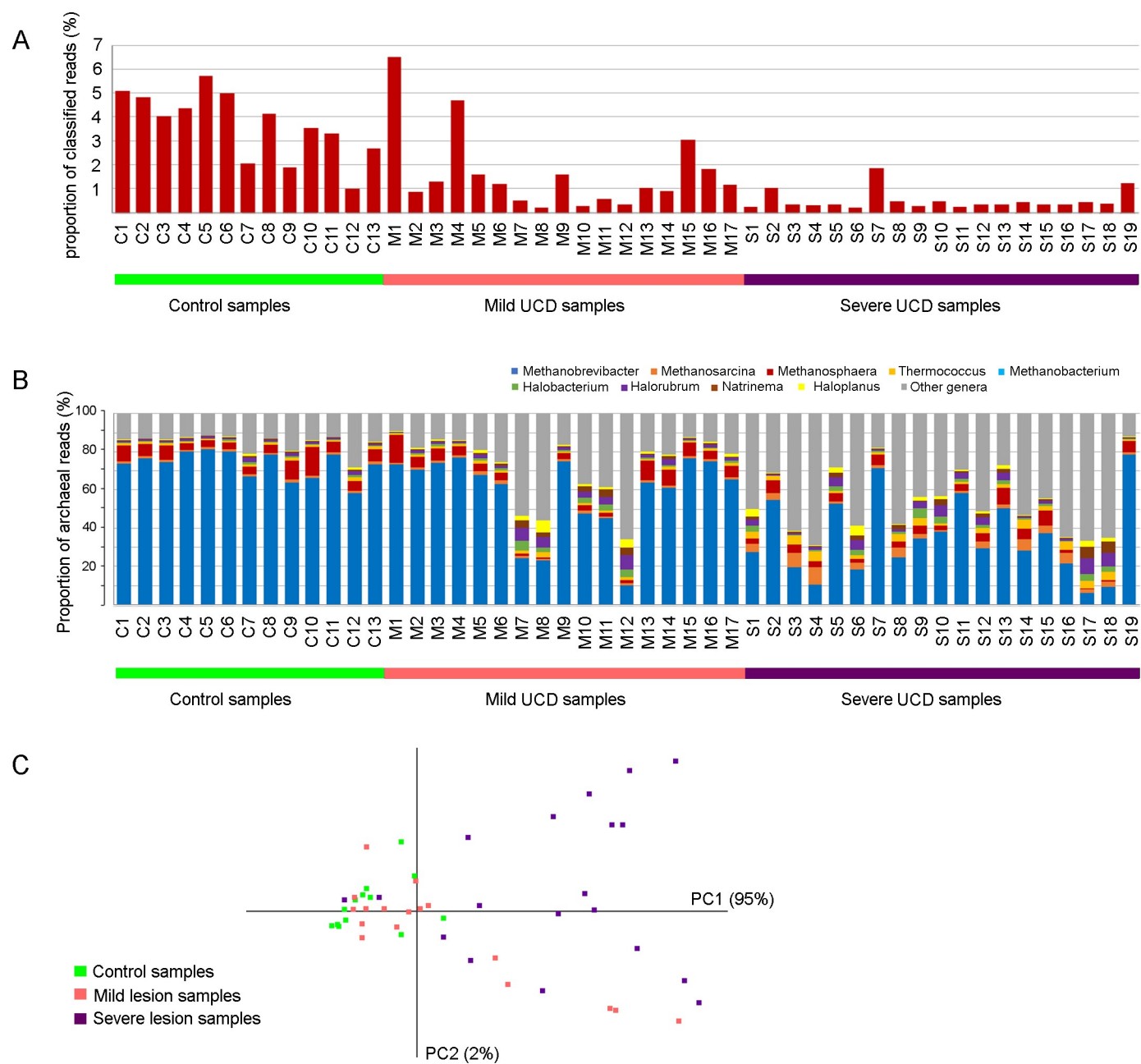

**Fig 6. Archaeal abundance.** The archaeal proportion of all classified reads within each sample (A), the distribution of archaeal genera representing ≥5% of the archaeal reads in at least one sample (B) and a principal component analysis (C), based on samples from mild (n = 17) and severe (n = 19) udder cleft dermatitis (UCD) and samples from skin at the fore udder attachment from healthy controls (n = 13).

**Fungi.** The fungal reads represented a mean of 0.5 (SD 0.4), 2.6 (SD 9.2) and 0.2 (SD 0.1)% of the classified reads in control samples, mild UCD, and severe UCD, respectively. Control samples had a higher proportion of fungal reads, compared to mild ($P = 0.05$) and severe ($P = 0.001$), but we found no indication of any specific genera and species that differed between sample types. Fusarium was the most frequently classified genus with a mean of 12,5 (SD 3.2), 10.3 (SD 4.2) and 10.2 (SD 2.3)% of the fungal reads for control, mild and severe samples, respectively. One sample was responsible for the numerically higher mean abundance in

samples from mild lesions (Fig 2A). In this specific sample, the majority of the classified fungal reads belonged to the genus *Candida*, with the species *Candida orthopsilosis* representing 69% of the fungal reads.

## Differences in microbiota between different time points–two examples

In two cases, the data included samples of the same UCD lesion from two different time points. In the first case, the sample (M7) was first identified in December 2018 as a recently developed mild lesion (Fig 7A). Six weeks later, the lesion was scored as a recently developed severe UCD lesion and was therefore sampled again (sample S10; Fig 7B). In both samples, bacteria

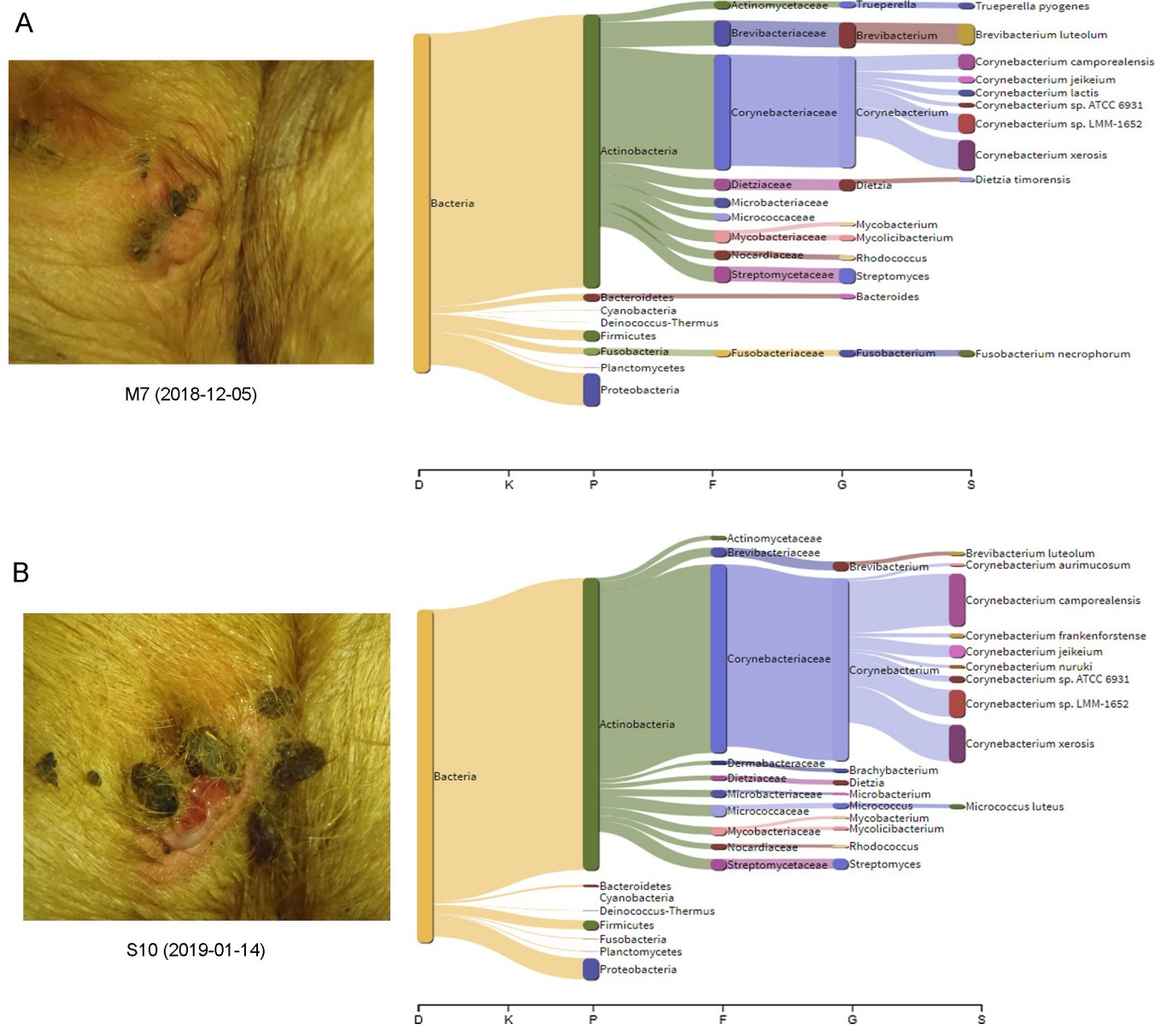

**Fig 7. Example 1.** Sankey visualization (obtained by the metagenomics tool Pavian) of the microbiota in a mild (A) and a severe (B) udder cleft dermatitis lesion in the same cow, sampled in December 2018 and six weeks later in January 2019. The flow diagram illustrates the proportion of bacterial reads assigned to a specific taxon at domain (D), kingdom (K), phylum (P), family (F), genera (G) and species level.

constituted more than 99% of the classified reads and the proportion of archaeal reads was low. Both samples had a high relative abundance of Actinobacteria and *Corynebacterium* (Fig 7). However, in the severe UCD sample S10, the proportion of *Corynebacterium* spp. had increased (60.5 of classified reads) compared to the initial mild UCD sample M7 (37.3% of classified reads), with *Corynebacterium camporealensis* being mainly responsible for the difference (4.6% of classified reads in M7 compared to 16.3% in S10) (Fig 7). The species level Shannon diversity index was also numerically higher in sample M7 (5.7) compared to sample S10 (4.9). Thus, the microbiota in this UCD lesion shifted over time towards a lesion containing mainly *Corynebacterium*.

In the second case, a sample (S6) was taken from a cow first identified as having a recently developed severe UCD lesion (Fig 8A), and the cow was then sampled again, around three months later at the final herd visit (sample S7; Fig 8B). As was seen in the previous example, bacteria constituted most of the classified reads, 99.6% of S6 and 97.6% of S7. Among the bacteria, *Brevibacterium luteolum* was most prevalent in S6, representing 49.2 of the classified reads, but only 0.1% in S7. *Corynebacterium* was common in both samples, representing 23.7 and 38.0% in S6 and S7, respectively, with the most abundant species identified as *Corynebacterium lactis* in S6 (6.4% compared to 1.9% in S7) and *Corynebacterium urealyticum* in S7 (14.9% compared to 0.2% in S6). Apart from these differences, *Porphyromonas asaccharolytica* (0.1 and 5.0% in S6 and S7, respectively) and *Fusobacterium necrophorum* (0.6 and 2.7% in S6 and S7, respectively) had increased in S7 compared to S6. Thus, the microbiota in this lesion shifted towards a higher proportion of anaerobic bacteria, and the species level Shannon diversity index shifted from 3.5 in S6 to 5.9 in S7.

## Discussion

To our knowledge, this is the first study to investigate the microbiota of UCD lesions using a shotgun metagenomic sequencing approach, and it provides an increased understanding of the microbiological differences in mild and severe UCD lesions compared to healthy skin. Our results show an altered microbiota in both mild and severe UCD lesions compared to the control samples, manifested by decreased bacterial diversity and an increased proportion of certain bacterial genera and species. In line with our results on diversity, Sorge et al. [17] found a lower bacterial diversity in samples from UCD lesions compared to control samples. A similar finding was also reported in a study on digital dermatitis, in which bacterial diversity decreased as the digital dermatitis lesions progressed in severity [32]. In metagenomic studies of the human skin, a high diversity is characteristic of a healthy skin microbiota [33], whereas a loss of diversity and increased proportion of pathogenic or opportunistic bacteria can be defined as dysbiosis [33, 34]. Decreased diversity and dysbiosis are associated with numerous skin conditions, such as atopic dermatitis in humans and dogs [35, 36]. A lower diversity has also been found in diabetic foot ulcers and in the healthy skin of diabetic patients compared to non-diabetic controls [37]. However, it has not been established whether the dysbiosis is a cause or a result of the pathological condition, and further studies within this area are required. We also found that a few of the UCD samples, mainly from mild UCD, had similar microbiota to control samples, indicating that an altered microbiota is not always present in mild lesions.

### Bacterial abundance and subgroups of UCD samples

Apart from the decreased diversity, we found an increased proportion of certain bacterial genera and species in the UCD samples compared to the controls. Noticeably, it was not always the same species that increased in different samples. In our dataset, we observed four broad subgroups of UCD samples with different types of dysbiosis. In the largest subgroup, the genus

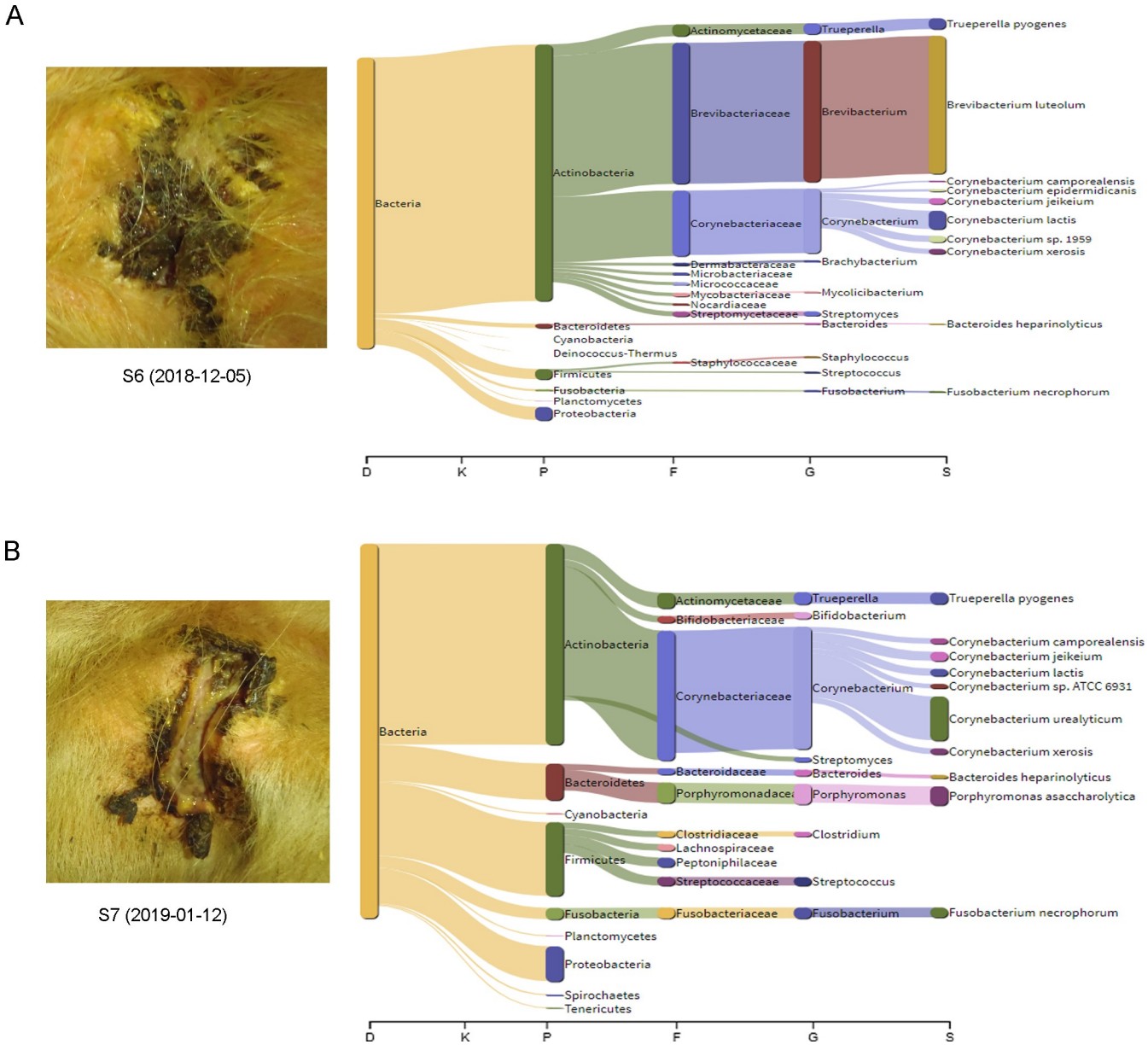

**Fig 8. Example 2.** Sankey visualization (obtained by the metagenomics tool Pavian) of the microbiota of a severe udder cleft dermatitis (UCD) lesion sampled at two different time points from the same cow, in October 2018 (A) and in January 2019 (B). The flow diagram illustrates the proportion of bacterial reads assigned to a specific taxon at domain (D), kingdom (K), phylum (P), family (F), genera (G) and species level.

*Corynebacterium* contributed to a high proportion of reads, although differences in species were seen between samples. In another subgroup of samples, mainly from mild lesions, *Staphylococcus* spp. represented a relatively high proportion of the classified reads and contributed to the decreased diversity. Several staphylococcal species were found but, in most cases, only one or two staphylococcal species represented more than 1% of the reads within the same sample. Both *Corynebacterium* spp. and *Staphylococcus* spp. have previously been identified in UCD lesions through culturing [3, 13]. The genera *Corynebacterium* and *Staphylococcus* are also associated with bovine healthy skin from teat apices [38, 39] and, in contrast to our results, Sorge et al. [17] found a higher abundance of *Corynebacterium* spp. in controls than in UCD

samples. These bacteria are also common in human chronic ulcers, as well as being a common finding in healthy skin microbiota in humans [37]. *Staphylococcus* spp. are also associated with atopic dermatitis in both humans and dogs [23, 35] and are also known to be involved in biofilm formation in human chronic ulcers [40]. A few samples, (three mild and one severe UCD samples) had a high proportion of *Brevibacterium* (i.e. *Brevibacterium luteolum*), a coryneform bacteria that, to our knowledge, has not been previously identified in UCD lesions. *Brevibacterium* is a known skin commensal in humans that, in similarity to corynebacteria, may act as an opportunist, mainly in immunocompromised individuals [41, 42]. The high proportion of common skin bacteria found in UCD samples in the present study implies that, under certain circumstances, skin commensals may increase in relative abundance, resulting in a decreased diversity indicative of an impaired microbiota, or dysbiosis. Such circumstances may include an altered local environment, for example, changes in pH, oxygen levels or humidity. The presence of a dysbiotic microbiota involving common skin bacteria is also seen in human chronic wounds for which there are underlying causes, such as diabetes or venous insufficiencies [24, 37]. In the fourth subgroup indicating dysbiosis, including several severe, but no mild, UCD samples, a more anaerobic microbiota was observed, including bacterial species such as *Trueperella pyogenes*, *Fusobacterium necrophorum* and *Porphyromonas asaccharolytica*. These bacteria have previously been identified in bacteriological studies of UCD [2, 3] and, in line with our results, Sorge et al. [17] found a higher abundance of these bacteria in UCD samples compared to control samples. Finding these opportunistic bacteria in severe UCD lesions is not surprising as they are associated with several bacterial conditions in ruminants, such as interdigital phlegmon, abscesses and wound infections [43, 44]. In chronic wounds, the local environment is associated with low oxygen levels that enable the growth of these bacteria [40]. In addition, *Fusobacteria* and *Porphyromonas* are commonly found in combination and, in a laboratory setting, have been shown to form biofilm and could therefore impair the healing of wounds [45]. In line with this, the severe sample (S7) that was re-sampled after a wound duration of around three months showed an increased proportion of these two bacteria. The clinical appearance of the wound that was sampled after three months also showed more characteristics associated with human chronic wounds, such as a lack of granulation tissue, presence of necrotic tissue and fibrin [46], compared to when it had recently developed. The importance of the higher abundance of *Bifidobacterium* spp. found in control samples is not known, but it could be speculated that these bacteria have a protective effect on the skin barrier. Sorge et al. [17] also found *Bifidobacterium* spp. to be significantly more common in controls compared to UCD samples. Bifidobacteria produce lactic acid and are commonly used as a probiotic treatment in the restoration of gut microbiota [47]. Although previous studies have suggested an association between UCD and *Treponema* spp. [1, 15], we found no evidence of *Treponema* spp. being involved in UCD lesions, as the abundance of this genus was low in all samples, which was also seen in a previous study of UCD [17].

### Presence of mastitis-causing bacteria

A low abundance of the bacteria considered to be important mastitis-causing pathogens in Sweden (apart from *Trueperella pyogenes*) was seen. Thus, our results do not indicate that UCD lesions act as a reservoir for such pathogens. *Trueperella pyogenes* is a mastitis-causing pathogen [44], typically infecting heifers and cows in the dry period during the summer. In such "summer mastitis", it is generally believed that flies are responsible for the transmission of the pathogen. This opportunistic bacterium is also common in several other conditions, such as abscesses and pneumonia [44], and we do not believe that the presence of *Trueperella pyogenes* in severe UCD lesions has any significant effect on the risk of mastitis in dairy herds.

However, during the summer, an increased presence of the bacteria in cows with severe UCD could increase the risk of this specific type of summer mastitis if flies are present in the environment. It is not clear, however, whether such a route of transmission could partly explain the association between UCD and mastitis found in previous studies [4, 14].

## Abundance of other microorganisms

To our knowledge, the presence of Archaea on bovine skin has not been previously investigated. These single-celled microorganisms have previously been identified in different environments, including the bovine rumen [48] and the human gut and skin [49]. Methanogens, such as *Methanobrevibacter*, have been found to play a role in ruminal microbial metabolism, by using hydrogen for their growth, and reducing carbon dioxide to methane [48]. The presence of *Methanobrevibacter* in our samples could be the result of contamination from faeces or the environment, but it is also possible that these archaea are part of the bovine skin microbiota. Archaea have been proposed to play a role in ammonia metabolism in human skin [49] and it could be speculated that they might play a similar role in pH regulation of bovine skin, although this requires further studies. We found differences in archaeal abundance between controls and mild and severe UCD samples, but the importance of these differences is not known as archaeal function is still a poorly explored area of research. Our results suggest that fungi are not associated with UCD lesions in most cases, as the fungal reads represented a low proportion of the majority of samples. However, one mild UCD sample had a high proportion of fungal reads, which indicates that opportunistic fungi may be part of a shift towards a decreased diversity of the microbiota in UCD lesions. In this sample, one specific fungal species was responsible for 69%of the fungal reads. We found low proportions of viral and protozoan reads in all samples, which imply that these agents are not a common part of the UCD microbiota or healthy bovine skin. However, the methods used for preparing samples for DNA extraction might have affected the viral content of the samples, as small virus particles could have been lost prior to DNA extraction. An alternative methodology would probably be needed for specific analysis of the viral content of the samples. Another pathogen that has been associated with UCD is the mange mite, *Chorioptes bovis* [10]. As *Chorioptes bovis* is a common finding in Sweden, it would have been interesting to see whether DNA from this pathogen was present in our samples. As no genome sequence was available from the NCBI, no such analysis was performed. In addition, as several studies did not find any evidence of an involvement of mange mites in UCD [2, 5], we do not believe that they are of major interest.

## Final remarks

The methods used in this study, regarding both the sampling and laboratory procedures, as well as shotgun metagenomic sequencing, have been scarcely explored for bovine wound microbiota. Shotgun metagenomic sequencing has, however, been shown to enhance the detection of bacterial species compared to 16S amplicon sequencing, in which the selection of primers may affect the results [19, 50]. Contaminating DNA in laboratory reagents may influence the taxonomic classification of metagenomics results [51]. In this study, we did not include negative controls and this constitutes a limitation when interpreting the results. This is especially true when looking at low abundance groups. However, in this study we focus mainly on the most dominating taxa and their proportions. Our results are generally in line with a previous study using 16S amplicon sequencing to investigate UCD lesions in comparison with control samples [17], suggesting that these two studies have increased the understanding of the microbiota of UCD lesions in comparison to healthy skin at the fore udder attachment. In addition, our results demonstrate that mild UCD lesions also display a dysbiotic microbiota

and, in combination with the fact that mild UCD lesions often develop into severe lesions [14], this suggests that mild lesions should not be ignored.

## Conclusions

The results of this study indicate that the microbiota of UCD lesions is different to that of healthy skin from the same body site, with a dysbiosis manifested as reduced diversity and increased proportions of certain bacteria in mild and severe UCD lesion samples. It is, however, not known if the dysbiosis is a contributing cause to, or a result of, the UCD lesions, and these associations require further investigations. In this study we identified three broad categories of dysbiosis characterized by different groups of bacteria. Although several bacterial species were more frequently identified in the UCD samples, the overall interpretation is that no specific pathogen is involved in the development of UCD, as the bacteria differed between samples. We found no evidence of UCD lesions acting as a reservoir for mastitis-causing bacteria, as such bacteria were found in low proportions in most samples.

## Supporting information

**S1 Table. Sample overview.** Overview of samples subjected to shotgun metagenomic sequencing in a study of the microbiota of mild (M) and severe (S) udder cleft dermatitis (UCD) in comparison with samples from the same body site from healthy controls (C) in 47 cows in 6 Swedish dairy herds.
(XLSX)

**S2 Table. Raw classification data.** The results of the classification of the sequenced reads (after removal of sequences assigned to the *Bos Taurus* genome) presented in an Excel table as both clade read counts and normalized into a percentage of classified reads for each taxa.
(XLSX)

**S1 Fig. PCA loadings.**
(TIF)

**S2 Fig. PCA axis 4 on bacterial genus level.**
(TIF)

**S3 Fig. PCA on bacterial genus level colored by herd, breed, parity and extraction month.**
(TIF)

**S4 Fig. Hierarchical clustering of samples.**
(TIF)

## Acknowledgments

Sequencing was performed by the SNP&SEQ Technology Platform in Uppsala. The facility is part of the National Genomics Infrastructure (NGI) Sweden and Science for Life Laboratory. The SNP&SEQ Platform is also supported by the Swedish Research Council and the Knut and Alice Wallenberg Foundation. We would also like to thank Harri Ahola and Karin Ullman at the National Veterinary Institute, Uppsala, Sweden for their help and advice regarding DNA extraction methodology, as well as Maria "Maja" Persson and the staff of "DOA lab" at the National Veterinary Institute for excellent technical assistance. Last but not least, the farmers and cows participating in this study–thank you for your kind hospitality and assistance throughout the study period.

## Author Contributions

**Conceptualization:** Lisa Ekman, Elisabeth Bagge, Ann Nyman, Karin Persson Waller, Märit Pringle, Bo Segerman.

**Data curation:** Lisa Ekman, Bo Segerman.

**Formal analysis:** Lisa Ekman, Bo Segerman.

**Funding acquisition:** Karin Persson Waller.

**Investigation:** Lisa Ekman, Ann Nyman, Karin Persson Waller.

**Methodology:** Lisa Ekman, Elisabeth Bagge, Ann Nyman, Märit Pringle, Bo Segerman.

**Project administration:** Lisa Ekman, Karin Persson Waller.

**Software:** Bo Segerman.

**Supervision:** Ann Nyman, Karin Persson Waller, Bo Segerman.

**Visualization:** Lisa Ekman, Bo Segerman.

**Writing – original draft:** Lisa Ekman.

**Writing – review & editing:** Elisabeth Bagge, Ann Nyman, Karin Persson Waller, Märit Pringle, Bo Segerman.

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
