## [Decision Letter · Decision Letter 0]

25 Aug 2020

PONE-D-20-18567

Udder cleft dermatitis in dairy cows is associated with an altered and less diverse microbiota compared to the microbiota of healthy skin

PLOS ONE

Dear Dr. Ekman,

Thank you for submitting your manuscript to PLOS ONE. After careful consideration, we feel that it has merit but does not fully meet PLOS ONE’s publication criteria as it currently stands. Therefore, we invite you to submit a revised version of the manuscript that addresses the points raised during the review process.

We look forward to receiving your revised manuscript.

Kind regards,

Peter Gyarmati

Academic Editor

PLOS ONE

Journal Requirements:

2. We note that you are reporting an analysis of a microarray, next-generation sequencing, or deep sequencing data set. PLOS requires that authors comply with field-specific standards for preparation, recording, and deposition of data in repositories appropriate to their field. Please upload these data to a stable, public repository (such as ArrayExpress, Gene Expression Omnibus (GEO), DNA Data Bank of Japan (DDBJ), NCBI GenBank, NCBI Sequence Read Archive, or EMBL Nucleotide Sequence Database (ENA)). In your revised cover letter, please provide the relevant accession numbers that may be used to access these data. For a full list of recommended repositories, see http://journals.plos.org/plosone/s/data-availability#loc-omics or http://journals.plos.org/plosone/s/data-availability#loc-sequencing.

We note that one or more of the authors are employed by a commercial company: Växa Sverige.

3.1. Please provide an amended Funding Statement declaring this commercial affiliation, as well as a statement regarding the Role of Funders in your study. If the funding organization did not play a role in the study design, data collection and analysis, decision to publish, or preparation of the manuscript and only provided financial support in the form of authors' salaries and/or research materials, please review your statements relating to the author contributions, and ensure you have specifically and accurately indicated the role(s) that these authors had in your study. You can update author roles in the Author Contributions section of the online submission form.

3.2. Please also provide an updated Competing Interests Statement declaring this commercial affiliation along with any other relevant declarations relating to employment, consultancy, patents, products in development, or marketed products, etc.  

4. We noted in your submission details that a portion of your manuscript may have been presented or published elsewhere.

"The manuscript is a part of a doctoral thesis and is thus published as a manuscript in the print version of the thesis (ISBN 978-91-7760-564-5). The thesis is available at https://pub.epsilon.slu.se/16894/, this manuscript is not included in the online publication. "

Please clarify whether this publication was peer-reviewed and formally published. If this work was previously peer-reviewed and published, in the cover letter please provide the reason that this work does not constitute dual publication and should be included in the current manuscript.

Reviewers' comments:

Reviewer's Responses to Questions

**Comments to the Author**

1. Is the manuscript technically sound, and do the data support the conclusions?

Reviewer #1: Partly

Reviewer #2: Partly

2. Has the statistical analysis been performed appropriately and rigorously? 

Reviewer #1: No

Reviewer #2: No

3. Have the authors made all data underlying the findings in their manuscript fully available?

Reviewer #1: Yes

Reviewer #2: Yes

4. Is the manuscript presented in an intelligible fashion and written in standard English?

Reviewer #1: Yes

Reviewer #2: Yes

5. Review Comments to the Author

Reviewer #1: Ekman et al. investigate the skin microbiome of healthy and diseased cows using a shotgun metagenomic sequencing approach. Although I believe the study to have merit, it does contain a number of methodological flaws that make it difficult to evaluate in its current form.

Major

DNA extraction, library preparation and sequencing

Negative controls are an important component for any type of experiment where measurements are being made. The importance of these and their impact have been documented extensively throughout the microbiome literature (see https://bmcbiol.biomedcentral.com/articles/10.1186/s12915-014-0087-z). Because they were not included in this experiment, it is an obvious limitation that should be discussed, but I also think that their impact depends on how you analyze and present results. For example, if you make claims about specific OTUs, then you should be confident that what you are describing is real, but if you make claims about general trends at a specific taxonomic level (e.g., phylum, genus, species, etc..), as you do here, then contamination may have less of an impact on the overall results, especially when DNA yield is high and contaminants are in low abundance. Please address the importance of negative controls in microbiome research and discuss the limitations of not including them in your study.

It is also not clear how putative batch effects were handled in this study. Were all samples extracted on the same day? Were they sequenced on the same date, on the same machine?

Bioinformatics

Overall, I think that the methods describing your bioinformatic methods is very well done; however, I do have a concern regarding the omission of how adapter sequences and low quality sequences were handled. Adapter contamination is known to affect alignment statistics for commonly used sequence aligners like BWA and Bowtie2 (https://academic.oup.com/bioinformatics/article/30/15/2114/2390096), resulting in reads that are incorrectly classified as “unmapped”. This does appear to have affected the reported proportions of “host” versus “non-host” DNA for severe UCD samples and others, as a large proportion of your sequence reads were classified to the Babesia bigemina genome, which I know to suffer from Bos taurus DNA contamination. Failure to remove these adapter sequences may have also affected classification statistics for other sequences processed with Kraken2. Please rerun your raw sequence data through a read trimming tool like Trimmomatic, cutadapt or other relevant tool before proceeding with the remainder of your analysis. Please also add a column to Supplementary Table 1 describing the number of sequence reads that survived this trimming and other quality control steps.

Statistics

Breed, herd, and parity are likely confounders that should be included in your statistical analysis. Please consider accounting for these potential confounders and update your methods.

Minor

Title: Please consider changing the title, as this seems to be a study comparing the microbiome of healthy and diseased skin (i.e., UCD).

Line 66: Sequencing methods (e.g., 16S amplicon sequencing, shotgun sequencing, etc..) are all biased in some way, as are the sequence databases used to describe the underlying results. You do an excellent job of discussing the tradeoffs of each approach later in the manuscript, but I recommend removing this sentence as it is inaccurate.

Line 77: Shotgun metagenomics can also reveal strain-level information. Please add this and a relevant citation.

Line 102: Change “All scoring and sampling were performed by the first author” to “All scoring and sampling was performed by a single researcher”.

Line 119: Was a pre-dipping solution applied prior to milking? If so, please add and specify relevant product information.

Line 215: Please clarify what you mean by “complexity”. What does it mean for the complexity of a sample to be estimated and why was this done?

Line 220: Please list the bacterial species that were included in this analysis.

Line 248: Please clarify what is meant by “... there was no convincing trend …” Was this based on a pre-determined p-value or “eyeball method”?

Line 270: How are the samples labeled on the ordination plots? It seems that some of the points/samples have labels, but others do not. What is the criterion for labeling samples?

Lines 476-477: The design of this study was probably not sufficient to determine “causation” of dysbiosis; consider re-wording this sentence and others (e.g., 567-569).

Line 555: Delete “Unbiased”. I know what you mean here, but there are numerous other sources of bias in microbiome studies. The choice of sampling device, the choice of DNA extraction kit, the choice of DNA shearing method (e.g., enzymatic versus mechanical), the choice of sequencing depth (e.g., shallow versus deep), and the choice of reference sequence database can all bias shotgun metagenomic experiments.

Reviewer #2: The primary goal of this manuscript is to compare the cutaneous microbiome of sites with mild/severe dermatitis lesions to healthy skin of the fore udder attachment. To accomplish this, the authors generated shotgun metagenomic sequencing data for 49 samples. Analysis of these samples with the k-mer approach Kraken revealed differential communities between controls and samples with UCD. Notably, subgroups existed within the UCD samples and few known mastitis-causing pathogens were identified. Refreshingly, they looked across kingdoms and not only at bacteria. While authors have generated a valuable dataset and the overall methodology is technically sound, there are a few areas that could be improved.

Methods:

- The authors utilized Kraken to classify their metagenomic sequencing reads and then directly used these results to draw conclusions about which genera and species were differential between groups. Because Kraken classifies individual reads to their best matching location in the taxonomic tree and does not actually estimate the abundance of species, the program Bracken (https://ccb.jhu.edu/software/bracken/) should be applied to the Kraken results and then the Bracken species abundances should be compared between groups.

Figure 3:

- Are any of the UCD subgroups, driven by the herd of the animal?

- In the "Bacterial abundance" section, the authors discuss several of the taxa that are driving these ordinations. Can the authors add biplot lines for these driver taxa to the PCA plots?

-For the species based results (Fig 3C), are similar conclusions drawn if the authors perform ordination analysis on a distance matrix, calculated with Bray Curtis, on the samples? It's unclear how there is so little variance at the species level between the control samples.

Figure 4:

- The authors frequently describe different UCD subgroups. To complement these descriptions, can the authors utilize a hierarchical clustering based method to more robustly define them?

Minor comments:

- In Fig 2A, the red/green coloring of Archaea and Fungi will be difficult for colorblind individuals to distinguish. Can the authors please update?

- To make Fig 2B more informative, can the authors use individual scales for the different kingdoms and denote p-values for those comparisons that are significant?

6. PLOS authors have the option to publish the peer review history of their article (what does this mean?). If published, this will include your full peer review and any attached files.

Reviewer #1: No

Reviewer #2: **Yes: **A Byrd

---

## [Author Response · Author response to Decision Letter 0]

9 Oct 2020

Response to editor comments:

1. The revised manuscript, including figures and supplemental material are updated according to the style requirements and templates.

2. The raw sequence data can be found in the NCBI Sequence Read Archive, with the accession number series SRR11913436–SRR11913484. 

3. 3.1. K.P.W. received funding from The Swedish Research council Formas (grant number 221 - 2013-269, www.formas.se) and from Stiftelsen lantbruksforskning - Swedish farmers' foundation for agricultural research (grant number V1430006, www.lantbruksforskning.se). The funders had no role in study design, data collection and analysis, decision to publish, or preparation of the manuscript. A.N. is employed by Växa Sverige, but her salary costs for her work in the study was covered by the grants listed above. Thus, Växa Sverige did not have any role in the study design, data collection and analysis, decision to publish, or preparation of the manuscript. The specific roles of these authors are articulated in the ‘author contributions’ section. 

3.2. The authors have declared that no competing interests exist. A.N. is employed by the commercial company Växa Sverige. This does not alter our adherence to PLOS ONE policies on sharing data and materials.

4. The manuscript has been published as a part of a doctoral thesis: However, this publication is not peer reviewed, and it does not constitute dual publication. 

5. We also discovered an error in the Ethics Statement, as an older approval number was used by mistake. The correct statement should read: Ethical approval was obtained from a regional Swedish ethics committee appointed by the Swedish board of Agriculture, approval number 5.8.18-06335/2018. Also updated in the box during the submission of the revision. 

Response to reviewers:

Reviewer #1: Ekman et al. investigate the skin microbiome of healthy and diseased cows using a shotgun metagenomic sequencing approach. Although I believe the study to have merit, it does contain a number of methodological flaws that make it difficult to evaluate in its current form.

Major

DNA extraction, library preparation and sequencing

Negative controls are an important component for any type of experiment where measurements are being made. The importance of these and their impact have been documented extensively throughout the microbiome literature (see https://bmcbiol.biomedcentral.com/articles/10.1186/s12915-014-0087-z). Because they were not included in this experiment, it is an obvious limitation that should be discussed, but I also think that their impact depends on how you analyze and present results. For example, if you make claims about specific OTUs, then you should be confident that what you are describing is real, but if you make claims about general trends at a specific taxonomic level (e.g., phylum, genus, species, etc..), as you do here, then contamination may have less of an impact on the overall results, especially when DNA yield is high and contaminants are in low abundance. Please address the importance of negative controls in microbiome research and discuss the limitations of not including them in your study.

We have added a short discussion on this limitation in the final remarks. 

It is also not clear how putative batch effects were handled in this study. Were all samples extracted on the same day? Were they sequenced on the same date, on the same machine?

We have added information about extraction day in S1 Table. All sequencing was done in a single novaseq-run. To roughly determine that DNA extraction date did not interfer with the identified subgroups of samples we colored the samples by extraction month (which is also related to sampling date) in the PCA plot for the bacterial genus level and found no evidence of such interference (Fig S3).

Bioinformatics

Overall, I think that the methods describing your bioinformatic methods is very well done; however, I do have a concern regarding the omission of how adapter sequences and low quality sequences were handled. Adapter contamination is known to affect alignment statistics for commonly used sequence aligners like BWA and Bowtie2 (https://academic.oup.com/bioinformatics/article/30/15/2114/2390096), resulting in reads that are incorrectly classified as “unmapped”. This does appear to have affected the reported proportions of “host” versus “non-host” DNA for severe UCD samples and others, as a large proportion of your sequence reads were classified to the Babesia bigemina genome, which I know to suffer from Bos taurus DNA contamination. Failure to remove these adapter sequences may have also affected classification statistics for other sequences processed with Kraken2. Please rerun your raw sequence data through a read trimming tool like Trimmomatic, cutadapt or other relevant tool before proceeding with the remainder of your analysis. Please also add a column to Supplementary Table 1 describing the number of sequence reads that survived this trimming and other quality control steps.

Indeed, there was a fraction of the reads that had adaptor content. We have run all fastq files through trimmomatic to remove them. There were very few reads dropped during this procedure, but up to ~15% of the reads were shortened by the trimming. The kraken analysis was then updated. We used a new fresh build of the kraken database in this step and added a Bracken analysis. We have updated the bioinformatic description in the method section and the information in table S1. 

Statistics

Breed, herd, and parity are likely confounders that should be included in your statistical analysis. Please consider accounting for these potential confounders and update your methods.

We agree that especially herd, but also breed and parity, may have an effect on the microbiota. To investigate the distribution of these factors over the control and UCD groups we performed Fisher’s exact test and found that these factors did not differ significantly between the groups (this information has been updated in the manuscript). As we wanted to compare the bacterial abundance between groups and as the number of observations were limited (and the bacteria far from normally distributed within samples), we believe that a multivariable analysis is less useful for this dataset. As a spot check, we did test a mixed effect ordinal logistic regression model with bacterial abundance as an ordinal outcome for analyzing some of the most abundant species and found that the estimates (and significance levels) for UCD status was largely unaffected when these potential confounders were included (although herd was significantly associated with e.g. Bifidobacterium abundance). Thus, we would like to keep the Wilcoxon rank sum tests in the manuscript.

In addition, to visually interpret if any of the above mentioned variables had any substantial impact on the clustering of samples in the PCA, we colored the samples by herd, breed and parity in the PCA plot for the bacterial genus level, and found no evidence of such associations (FigS3). 

Minor

Title: Please consider changing the title, as this seems to be a study comparing the microbiome of healthy and diseased skin (i.e., UCD).

We have changed the title.

Line 66: Sequencing methods (e.g., 16S amplicon sequencing, shotgun sequencing, etc..) are all biased in some way, as are the sequence databases used to describe the underlying results. You do an excellent job of discussing the tradeoffs of each approach later in the manuscript, but I recommend removing this sentence as it is inaccurate.

We have removed this sentence.

Line 77: Shotgun metagenomics can also reveal strain-level information. Please add this and a relevant citation.

We have added this and included a recent review on this topic to the citations.

Line 102: Change “All scoring and sampling were performed by the first author” to “All scoring and sampling was performed by a single researcher”.

Changed according to suggestion.

Line 119: Was a pre-dipping solution applied prior to milking? If so, please add and specify relevant product information.

The majority of herds did not use any pre-dipping solution, although all of them used some kind of teat spray or dip post-milking. We do not have any information on the products that were used, but since the UCD lesions are located rather far from the teats, and as all cows within a herd (controls and UCD cows) received the exact same products and procedures during milking, we do not believe that this had any effect on the results of the study.

Line 215: Please clarify what you mean by “complexity”. What does it mean for the complexity of a sample to be estimated and why was this done?

We agree that this analysis is somewhat difficult to interpret. We have removed this from the manuscript (including Fig 5C).

Line 220: Please list the bacterial species that were included in this analysis.

We have indicated these bacterial species in the S2 Table (Bracken results presented in percentages). 

Line 248: Please clarify what is meant by “... there was no convincing trend …” Was this based on a pre-determined p-value or “eyeball method”?

We have updated the information of fungal reads in the manuscript. There were no significant differences between sample types in the proportion of classified reads for protozoa and viruses, we have now added this information in Figure 2. In addition, a brief “eyeball method” of the table of classified viral and protozoan reads gave no indication that any taxa were more abundant in any specific sample type.

Line 270: How are the samples labeled on the ordination plots? It seems that some of the points/samples have labels, but others do not. What is the criterion for labeling samples?

To make a more unbiased presentation of the data, all samples have now been indicated in the plots.

Lines 476-477: The design of this study was probably not sufficient to determine “causation” of dysbiosis; consider re-wording this sentence and others (e.g., 567-569).

We have changed the wording in this sentence to clarify. In our understanding, a decreased diversity is indicative of an unbalanced microbiota, and thus a dysbiosis, so it was not our intention to assume any causation. We also added a sentence in the Conclusions section to clarify this.

Line 555: Delete “Unbiased”. I know what you mean here, but there are numerous other sources of bias in microbiome studies. The choice of sampling device, the choice of DNA extraction kit, the choice of DNA shearing method (e.g., enzymatic versus mechanical), the choice of sequencing depth (e.g., shallow versus deep), and the choice of reference sequence database can all bias shotgun metagenomic experiments.

We have deleted the word unbiased.

Reviewer #2: The primary goal of this manuscript is to compare the cutaneous microbiome of sites with mild/severe dermatitis lesions to healthy skin of the fore udder attachment. To accomplish this, the authors generated shotgun metagenomic sequencing data for 49 samples. Analysis of these samples with the k-mer approach Kraken revealed differential communities between controls and samples with UCD. Notably, subgroups existed within the UCD samples and few known mastitis-causing pathogens were identified. Refreshingly, they looked across kingdoms and not only at bacteria. While authors have generated a valuable dataset and the overall methodology is technically sound, there are a few areas that could be improved.

Methods:

- The authors utilized Kraken to classify their metagenomic sequencing reads and then directly used these results to draw conclusions about which genera and species were differential between groups. Because Kraken classifies individual reads to their best matching location in the taxonomic tree and does not actually estimate the abundance of species, the program Bracken (https://ccb.jhu.edu/software/bracken/) should be applied to the Kraken results and then the Bracken species abundances should be compared between groups.

We have run the data through Bracken to estimate species, genera and phylum level data. We have updated the manuscript to use Bracken data instead of Kraken data. We had to build a new Kraken database and reanalyze the data, since the previously used database lacked files required by Bracken. 

Figure 3:

- Are any of the UCD subgroups, driven by the herd of the animal?

We agree that herd (and also breed and parity), may have an effect on the microbiota. To visually interpret if any of the these variables had any substantial impact on the clustering of samples in the PCA, we colored the samples by herd, breed and parity in the PCA plot for the bacterial genus level, and found no evidence of such associations (FigS3). 

In addition, to investigate the distribution of herd, breed and parity over the control and UCD groups we performed Fisher’s exact test and found that these factors did not differ significantly between the groups (this information has been updated in the manuscript). As we wanted to compare the bacterial abundance between groups and as the number of observations were limited (and the bacteria far from normally distributed within samples), we believe that a multivariable analysis is less useful for this dataset. As a spot check, we did test a mixed effect ordinal logistic regression model with bacterial abundance as an ordinal outcome for analyzing some of the most abundant species, with herd as random factor, and found that the estimates (and significance levels) for UCD status was largely unaffected when these potential confounders were included (although herd was significantly associated with e.g. Bifidobacterium abundance). 

- In the "Bacterial abundance" section, the authors discuss several of the taxa that are driving these ordinations. Can the authors add biplot lines for these driver taxa to the PCA plots?

In our opinion the figure got too dense with biplot lines, but we have added a supplemental figure with loadings for the PC axis used (S2 Fig). 

-For the species based results (Fig 3C), are similar conclusions drawn if the authors perform ordination analysis on a distance matrix, calculated with Bray Curtis, on the samples? It's unclear how there is so little variance at the species level between the control samples.

The control samples did not show a large variability for the taxa which were the main drivers for the displayed PC axes. However, the control samples do spread out in projections with higher PCs. We do not have the Bray Curtis distance option in the PCA tool used. We have tried to plot a Bray Curtis based ordination in R. The control samples cluster in there as well, although not fully as tight as in the PCA, and UCD samples cluster in roughly similar patterns as in Fig 3C. 

Figure 4:

- The authors frequently describe different UCD subgroups. To complement these descriptions, can the authors utilize a hierarchical clustering based method to more robustly define them?

We have added a hierarchical clustering analysis to the supplement (S4 Fig). 

Minor comments:

- In Fig 2A, the red/green coloring of Archaea and Fungi will be difficult for colorblind individuals to distinguish. Can the authors please update?

We have changed the colors.

- To make Fig 2B more informative, can the authors use individual scales for the different kingdoms and denote p-values for those comparisons that are significant?

We have made an enlargement for the minor taxa and added the p-values.

---

## [Decision Letter · Decision Letter 1]

30 Oct 2020

PONE-D-20-18567R1

A shotgun metagenomic investigation of the microbiota of udder cleft dermatitis in comparison to healthy skin in dairy cows

PLOS ONE

Dear Dr. Ekman,

Thank you for submitting your manuscript to PLOS ONE. After careful consideration, we feel that it has merit but does not fully meet PLOS ONE’s publication criteria as it currently stands. Therefore, we invite you to submit a revised version of the manuscript that addresses the points raised during the review process.

We look forward to receiving your revised manuscript.

Kind regards,

Peter Gyarmati

Academic Editor

PLOS ONE

Reviewers' comments:

Reviewer's Responses to Questions

**Comments to the Author**

1. If the authors have adequately addressed your comments raised in a previous round of review and you feel that this manuscript is now acceptable for publication, you may indicate that here to bypass the “Comments to the Author” section, enter your conflict of interest statement in the “Confidential to Editor” section, and submit your "Accept" recommendation.

Reviewer #1: (No Response)

Reviewer #2: (No Response)

2. Is the manuscript technically sound, and do the data support the conclusions?

Reviewer #1: Yes

Reviewer #2: Yes

3. Has the statistical analysis been performed appropriately and rigorously? 

Reviewer #1: Yes

Reviewer #2: Yes

4. Have the authors made all data underlying the findings in their manuscript fully available?

Reviewer #1: Yes

Reviewer #2: Yes

5. Is the manuscript presented in an intelligible fashion and written in standard English?

Reviewer #1: Yes

Reviewer #2: Yes

6. Review Comments to the Author

Reviewer #1: Thank you for responding to our comments in the previous round. There are just a few minor issues noted on most recent version, which would improve the readability and interpretability of the manuscript:

Line 144: Add catalog numbers for this product and others where applicable.

Line 161: Change “Covairs E220” to “Covaris E220” and add manufacturer information as you do elsewhere.

Lines 181-182: Add in-text citation for Trimmomatic and in reference list. Change trimmomatics to “Trimmomatic” or “trimmomatic”.

Line 252: Remove sample identifiers from PCA plots in Figure 3 and elsewhere. This will make it easier for your readers to see clustering by groups.

Line 412 and 428: Provide a more detailed description of these visualizations, including what they are meant to convey, axes descriptions (e.g., D = Domain, K = Kingdom, etc..), and label descriptions (e.g., integer labels above each taxonomic unit denote the number of reads classified). Consider replacing integer labels above each taxonomic unit with relative abundances, as I think this would do a better job of highlighting changes in the microbiome between each time point.

Line 544: Specify the kind of contamination you are referring to (e.g., DNA).

Reviewer #2: I thank the authors for taking the time to address my concerns, particularly those around the analysis approach and the PCA plots. I do have several additional minor comments

- Fig S2: There should be a legend for the point colors.

- Fig S3: The axes should be labeled.

- Bioinformatics analyses section:

o trimmomatics should be Trimmomatic and appropriately cited

o Parameters used for Trimmomatic, Kraken, Bracken should be specified.

- Reference should be given for mastitis-causing pathogens

- Mastitis-causing bacteria and spirochetes in UCD samples section: Epidermidis should not be capitalized.

- Fig 6B: Boxplots similar to Fig2B would allow easier visual comparison between groups.

7. PLOS authors have the option to publish the peer review history of their article (what does this mean?). If published, this will include your full peer review and any attached files.

Reviewer #1: No

Reviewer #2: No

---

## [Author Response · Author response to Decision Letter 1]

9 Nov 2020

Reviewer #1: Thank you for responding to our comments in the previous round. There are just a few minor issues noted on most recent version, which would improve the readability and interpretability of the manuscript:

Line 144: Add catalog numbers for this product and others where applicable.

We have added this information where available. 

Line 161: Change “Covairs E220” to “Covaris E220” and add manufacturer information as you do elsewhere.

We have corrected spelling and added the manufacturer information. 

Lines 181-182: Add in-text citation for Trimmomatic and in reference list. Change trimmomatics to “Trimmomatic” or “trimmomatic”.

We have corrected the text and added a reference.

Line 252: Remove sample identifiers from PCA plots in Figure 3 and elsewhere. This will make it easier for your readers to see clustering by groups.

We have removed the sample identifiers from the PCA plots in Fig 3, 6C and S2. 

Line 412 and 428: Provide a more detailed description of these visualizations, including what they are meant to convey, axes descriptions (e.g., D = Domain, K = Kingdom, etc..), and label descriptions (e.g., integer labels above each taxonomic unit denote the number of reads classified). Consider replacing integer labels above each taxonomic unit with relative abundances, as I think this would do a better job of highlighting changes in the microbiome between each time point.

We have added a further description of Fig 7 & 8. We did consider changing the integer labels to relative abundance as we agree that that would be more of interest here. However, this was not possible within the Pavian visualization tool, and as we believe that the label text is on the verge of too small, we decided to remove the integer labels from the flow diagrams.

Line 544: Specify the kind of contamination you are referring to (e.g., DNA).

We have specified the type of contamination. 

Reviewer #2: I thank the authors for taking the time to address my concerns, particularly those around the analysis approach and the PCA plots. I do have several additional minor comments

- Fig S2: There should be a legend for the point colors.

- Fig S3: The axes should be labeled.

We have added this information in Fig S2 and Fig S3.

- Bioinformatics analyses section:

o trimmomatics should be Trimmomatic and appropriately cited

We have corrected this information. 

o Parameters used for Trimmomatic, Kraken, Bracken should be specified.

We have included the parameters used for these tools. 

- Reference should be given for mastitis-causing pathogens

We have added this information. 

- Mastitis-causing bacteria and spirochetes in UCD samples section: Epidermidis should not be capitalized.

We have corrected the text here.

- Fig 6B: Boxplots similar to Fig2B would allow easier visual comparison between groups.

As the proportion of archaeal reads differed a lot between samples, with a generally lower abundance in the UCD samples (especially the severe samples) compared to the controls we believe that box plots might lead to over-interpretation of differences between sample types that in reality might be more related to the amount, and thus we would like to keep Fig 6 in its current, more descriptive, form.

---

## [Editor Report · Decision Letter 2]

11 Nov 2020

A shotgun metagenomic investigation of the microbiota of udder cleft dermatitis in comparison to healthy skin in dairy cows

PONE-D-20-18567R2

Dear Dr. Ekman,

We’re pleased to inform you that your manuscript has been judged scientifically suitable for publication and will be formally accepted for publication once it meets all outstanding technical requirements.

Kind regards,

Peter Gyarmati

Academic Editor

PLOS ONE
---

## [Editor Report · Acceptance letter]

19 Nov 2020

PONE-D-20-18567R2 

A shotgun metagenomic investigation of the microbiota of udder cleft dermatitis in comparison to healthy skin in dairy cows 

Dear Dr. Ekman:

I'm pleased to inform you that your manuscript has been deemed suitable for publication in PLOS ONE. Congratulations! Your manuscript is now with our production department. 

Kind regards, 

on behalf of

Dr. Peter Gyarmati 

Academic Editor

PLOS ONE